# A predictive model of the effect of therapeutic radiation on the human ovary

**Thomas W. Kelsey** [1]*, **Chia-Ho Hua**[2], **Amber Wyatt**[3], **Danny Indelicato**[3], **W. Hamish Wallace**[4]

**1** School of Computer Science, University of St Andrews, St Andrews, United Kingdom, **2** Department of Radiation Oncology, St. Jude Children's Research Hospital, Memphis, Tennessee, United States of America, **3** Department of Radiation Oncology, University of Florida, Jacksonville, Florida, United States of America, **4** Department of Haematology/Oncology, Royal Hospital for Children and Young People, Edinburgh, United Kingdom

* twk@st-andrews.ac.uk

**Data Availability Statement:** The research data supporting this publication can be accessed at https://doi.org/10.17630/4fe0d6ac-f519-4cb5-af45-6d81bb9e845e.

## Abstract

Radiation to the female pelvis as part of treatment for cancer predisposes young women to develop Premature Ovarian Insufficiency (POI). As the human female is born with their full complement of non-growing follicles which decline in an exponential fashion until the menopause, the age at which POI occurs is dependent on the age of the patient at treatment and the dose received by the ovary. A model that predicts the age at which POI occurs for a known dose at a known age will aid counselling patients on their fertility risk. Patients deemed to be at high risk of POI may be considered to be good candidates for established fertility preservation techniques. An updated and externally validated model of the age-related decline in human ovarian reserve was combined with the best available estimate of the median lethal dose $LD_{50}$ for the human ovary. Using known age at diagnosis and posited radiotherapy treatment plan to estimate the dose to the least-affected ovary, we use an age-related model of the decline in ovarian reserve to generate a personalized age prediction of premature ovarian insufficiency. Our algorithm is available as an online calculator which graphs model outputs to inform discussions around survivor fertility. We report four example cases across different ages and diagnoses, each with two carefully designed photon and proton treatment plans. The treatment options are compared in terms of remaining fertile lifespan for the survivor. International oncology guidelines now mandate the consideration of later fertility when reviewing treatment options for children diagnosed with cancer. Our calculator (https://sites.cs.st-andrews.ac.uk/radiosensitivity), and the underlying algorithm and models, allow detailed predictions of the impact of various radiotherapy plans on fertility. These patient-specific data enhance pre-treatment discussions around post-treatment fertility and fertility preservation.

## Introduction

Continued advances in the management of children and young people with cancer have led to improved survival rates such that in excess of 80% of young people with cancer can expect to

**Funding:** The author(s) received no specific funding for this work.

**Competing interests:** The authors have declared that no competing interests exist.

be alive five years after diagnosis (83.8% for years 2008–2014 [1]). With improved outcomes, the quality of the survivorship has become increasingly important and the opportunity for fertility is a major goal [2]. In a population-based study in Scotland [3] on the effect of cancer and its treatment on subsequent pregnancy, women under age 40 at the time of diagnosis had a 38% overall reduction in the likelihood of pregnancy compared to the general population.

The ovaries may be damaged following total body, abdominal, or pelvic irradiation [4], and the extent of the damage is related to the radiation dose, fractionation schedule, and patient age at the time of treatment [5–7]. Premature ovarian insufficiency (POI) has been reported in 90% of patients after total body irradiation (TBI) (10.00–15.75 Gray (Gy)) and in 97% of females treated with whole abdominal irradiation (20–30 Gy) during childhood [8].

Human non-growing follicles (NGF) contain immature oocytes surrounded by flat, squamous granulosa cells (support cells) that are segregated from the oocyte's environment by the basal lamina. They are quiescent, showing little to no biological activity. They represent the human ovarian reserve. The NGF is sensitive to radiation, with an estimated $LD_{50}$ of less than 2 Gy [9]. This is the dose that will destroy roughly 50% of NGFs present in the ovary. In the absence of gonadotoxic chemotherapy, the number of NGFs present at the time of treatment, together with the radiation dose received by the ovaries, will influence the age when premature ovarian insufficiency (POI) occurs and determine the "fertile window".

Over the past few decades, the advent of proton radiotherapy has generated new possibilities for further reducing the radiation-induced toxicities but the optimal use of new technology hinges on precise dose-effect models to predict the benefit for individual patients. Compared to the modern alternative of intensity modulated photon therapy (IMRT), the difference is most marked in the reduction of low and intermediate dose exposure in children treated with proton therapy. This is particularly relevant in radiosensitive organs with a low $LD_{50}$, such as the ovaries. Proton therapy and other highly conformal radiation therapy have opened new possibilities for reduced late effects of treatment, but the optimal balance of target coverage and organ sparing is limited by imprecise dose-effect models.

Fertility preservation is now an established clinical area, with international oncology guidelines recommending that treatment options are discussed with respect to their long-term effects on fertility. Furthermore, newly diagnosed cancer patients at high risk of POI should, if there is sufficient time and they are well enough, be referred to reproductive specialists for discussion of fertility preservation options [10, 11]. Recent advances in fertility preservation offer new opportunities for young females that should be addressed before cancer treatment begins [12–15]. The aim of this study is to demonstrate how an assessment of the estimated radiation dose received by the ovary will allow the clinician to predict if an individual patient is likely to have a natural opportunity for fertility or is at risk of developing POI. Furthermore, assessing treatment plans from different techniques may allow choosing the best one that delivers the lowest radiation dose to the ovaries, thereby reducing the risk of developing POI, without compromising disease control. Fertility window prediction based on the estimated dose to the ovary furthest from the radiation target will allow clinicians to more accurately counsel women about their fertility prognosis and determine whether they may be a candidate for either established or experimental techniques of fertility preservation before their cancer treatment begins.

## Methods

The initial concept for a model for predicting the age at POI for a given chronologic age at treatment and a known radiation dose to the least-affected ovary was reported in [9, 16]. This model utilized a differential equation derived by Faddy and Gosden [17] for describing the

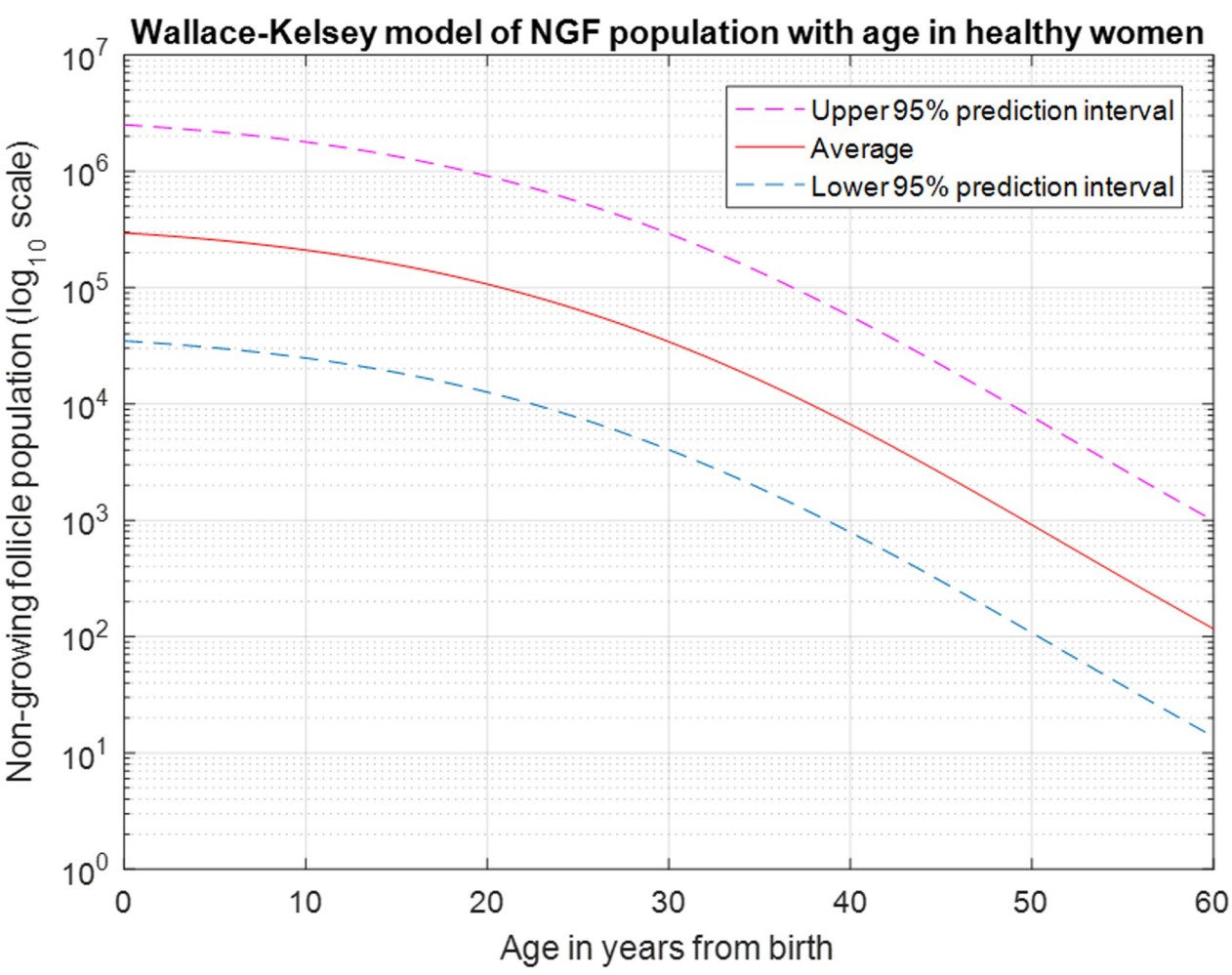

**Fig 1. The Wallace-Kelsey model of NGF decline from birth to age 60.** The average ovary contains approximately 300,000 NGFs at birth, declining to 100,000 at age 20, with about 800 remaining at average age of menopause (50–51 years).

natural follicle decline in healthy women, was considered the best available at that time, and was derived from a combination of published histologic data and the range of menopause age. In 2010, Wallace and Kelsey updated the model by fitting log-adjusted NGF population from 18 histologic studies (n = 325) with a 5-parameter asymmetric double-Gaussian cumulative curve. The equation with the best fit parameters is expressed as follows and is shown in Fig 1 for healthy women with average NGFs at birth and the 95% prediction interval to illustrate the individual variation.

$$log_{10}(NGF) = \frac{5.56}{4}\left[1 + Erf\left(\frac{age + 25.6 + \frac{52.7}{2}}{0.074\sqrt{2}}\right)\right]\left[1 + Erf\left(\frac{age - 25.6 - \frac{52.7}{2}}{24.5\sqrt{2}}\right)\right],$$

where *Erf* is the error function and *age* is the chronologic age.

Given the underlying NGF model and the estimate of the $LD_{50}$, the predictions have two levels of uncertainty: the dose received over a course of radiotherapy, and the z-score of the actual NGF population in standard deviations away from the predicted age-related average. Modern treatment planning systems for radiotherapy can calculate delivered ovary doses to a high accuracy if the ovaries are delineated on the treatment planning scan. Efforts can be made

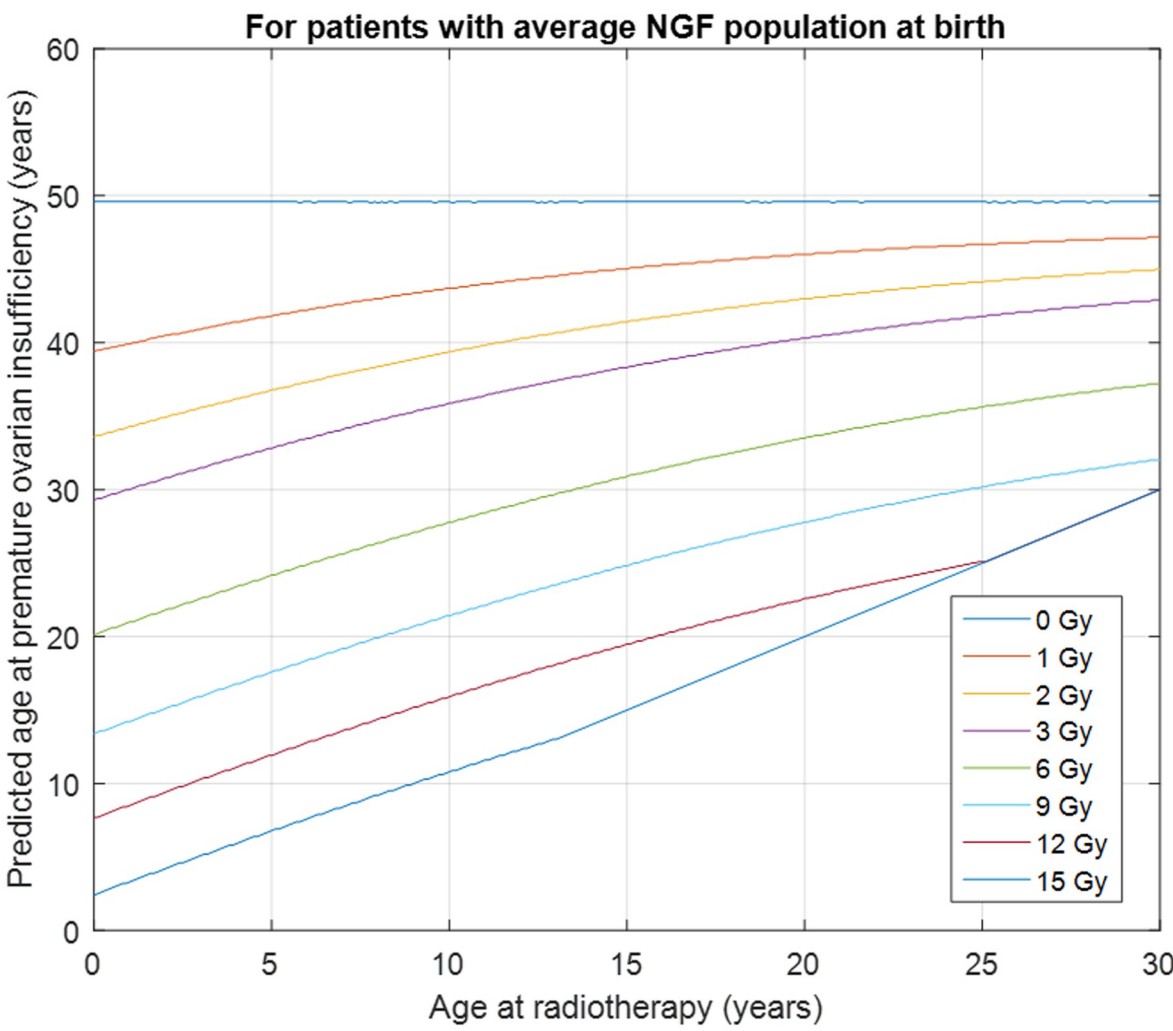

**Fig 2. Predicted age at POI for known age and known dose to the ovary.** A 5-year-old with average NGF population receiving 1 Gy to the ovary has predicted POI at age 42, losing about 8 years of reproductive lifespan; the same child treated with 12 Gy has predicted POI at age 12, indicating consideration of the use of fertility preservation techniques. The predictions are modifications of existing predictions [16] adapted to use the Wallace-Kelsey model of NGF population in healthy females [20].

to minimize the inter-fractional ovary movement, which may affect the total delivered dose, by maintaining a consistent bladder fullness throughout the radiotherapy course. There is currently no technology for determining NGF population in vivo, and no biomarkers for ovarian reserve have been validated for ages below 25 years [18, 19]. Therefore, our solution is to calculate a range of predicted ages at POI into which the majority of patients are expected to fall.

Visualization of the model prediction was achieved in two stages. Firstly, assuming that the patient has an average NGF population at the time of radiation therapy (i.e., that their age-related z-score is zero), we calculate the age at POI for the minimum, mean and maximum dose to the least-affected ovary, and overlay the results as a vertical prediction and range on Fig 2. Then, to take variation in NGF population into account, the treatment age is fixed and the minimum, mean, and maximum dose predictions are re-calculated at plus one and minus one standard deviation away from the mean age-related NGF population. The resulting prediction window accounts for 68% of the natural variation in NGF populations of the variation in age at menopause, and therefore applies to the majority of subjects. At both stages, age 30

years is plotted as an indicative threshold for changes of approach to the discussion of fertility preservation. The justification being that if you develop POI before 30 years of age you are unlikely to have a natural window of opportunity for natural fertility, as fertility declines markedly in the 10 years before the menopause.

Four representative pediatric patients who received radiotherapy were selected to illustrate an application of the predictive model. Both the "Common Rule" and the "Privacy Rule" (i.e.: HIPAA) allow the publication of a case report without consent or authorization from the subject provided the rules (listed at http://irb.ufl.edu/wp-content/uploads/Case-Reports.pdf) are adhered to.

The case reports used in this paper were reviewed by the University of Florida (UF) IRB, and found to be in compliance with the UF IRB and Privacy rules for case reports and thus neither IRB approval nor explicit consent is required.

The presented radiotherapy plans are anonymised and created as illustrative examples only. The purpose of the manuscript is to provide an algorithm to aid counselling for young women with cancer about their future fertility prospects. These case reports were chosen to sample a variety of ages that might illustrate variable ovary positions, ovarian volumes, and treatment techniques. These examples are not intended to suggest the superiority of any treatment modality. They included an 8-year-old child with pineoblastoma and a 19-year-old child with high risk medulloblastoma. Both were treated with 36 Gy craniospinal irradiation followed by a tumor boost to a total dose of 54 Gy. The other two illustrative patients were a 3-year-old and a 16-year-old each with a >8 cm unresectable pelvic Ewing sarcoma. These patients received an initial dose of 45 Gy followed by a tumor boost to a total dose of 64.8 Gy. The clinical target volume (CTV) for each case was consistent and designed according to contemporary disease-specific treatment protocols. The CTV was further expanded by 3 mm and 5 mm uniformly to form the cranial and extracranial planning target volume (PTV), respectively, with treatments delivered under daily image guidance.

The planning goal was to ensure that the tumor boost CTV and PTV were encompassed by >99% and >95% of the prescribed composite dose, respectively. Commercial treatment planning systems were used for the double-scattered 3D conformal proton plans (n = 3, Eclipse, Varian Medical Systems, Palo Alto, CA) and pencil beam intensity modulated proton plans (n = 1, RayStation, RaySearch Laboratories, Stockholm, Sweden). The double-scattered plans used 2–5 fields for the initial/CSI phase and 2–3 fields for the boost phase. The pencil beam plan used 3 fields for both the initial and boost phase. All patients were treated with proton therapy using a constant relative biological effectiveness of 1.1. For comparison, the pelvic sarcoma patients were replanned with 6 MV volumetric modulated arc photon therapy (VMAT) using 2 arcs for both the initial and boost phases. The CSI comparison photon plans were 3D conformal design using 3–4 fields for the CSI phase and 5 fields for the boost phase. All photon comparison plans were optimized in RayStation to ensure that the same PTV coverage criterion was achieved.

## Results

The updated Wallace-Kelsey model has been internally validated [20] and subsequently externally validated using data not used to derive the model [21]. The conservative estimate used for the $LD_{50}$ for NGFs is 2 Gy [9]. With this revised model, we can calculate the equivalent chronologic age corresponding to the remaining NGFs after irradiation and the menopause age when the total number of NGFs in the least-affected ovary drops below 1,000, assuming the NGF declining rate after irradiation follows that of a healthy woman at the same equivalent age. Fig 2 is an example which illustrates how this model can be used to predict the age at POI

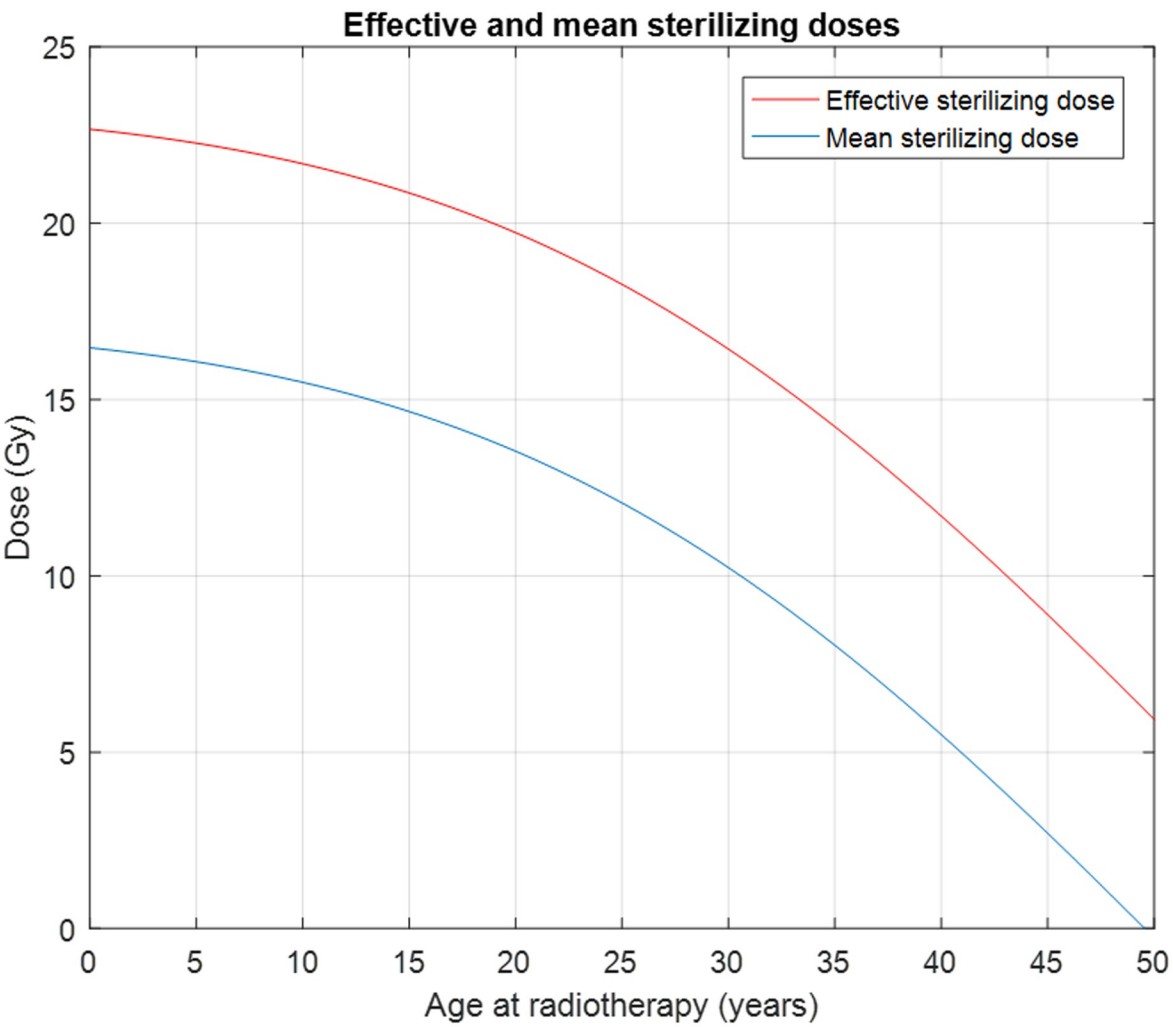

**Fig 3. The mean and effective sterilizing doses of radiation to the human ovary.** The blue line shows the radiation in Gy needed to sterilize 50% of subjects. The red line shows the radiation in Gy needed to sterilize 97.5% of subjects. The predictions are derived from the Wallace-Kelsey model of NGF population in healthy females [20], and the estimated radio sensitivity of the human NGFs [9].

for different radiation doses at various ages of radiation therapy for a female patient with the average NGF population for her chronological age. The model also allows the estimation of the Effective Sterilizing Dose (ESD) that will immediately reduce the NGF population below the level at natural ages at menopause (Fig 3).

The online tool for calculating ages at POI for supplied treatment details and known age is available at: https://sites.cs.st-andrews.ac.uk/radiosensitivity.

To use this online calculator, the user enters the age at radiotherapy of the patient and the planned mean, minimum and maximum dose in Gy for photon therapy or Cobalt Gray Equivalent (CGE) for proton therapy to the least affected ovary. The predicted age at POI is returned, together with lower and upper bounds for age at POI based on the supplied maximum and minimum ovary doses, respectively. The online tool also supplies this information in graphical form, with the predicted ages overlaid on the dose-response chart for the predictive algorithm (Fig 2). These charts can be compared for different treatment options and used

as the basis of discussions between the clinical team, the patient and their family, and reproductive specialists.

Four illustrative cases were considered (Figs 4a–4d, 5a–5d, 6a–6d & 7a–7d), each with photon and proton treatment plans. For each case, we compare the POI predictions on the same

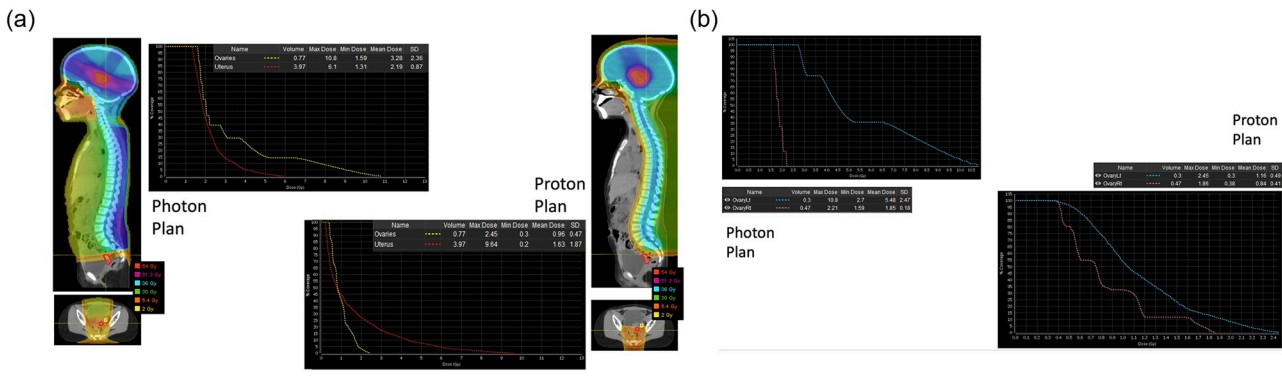

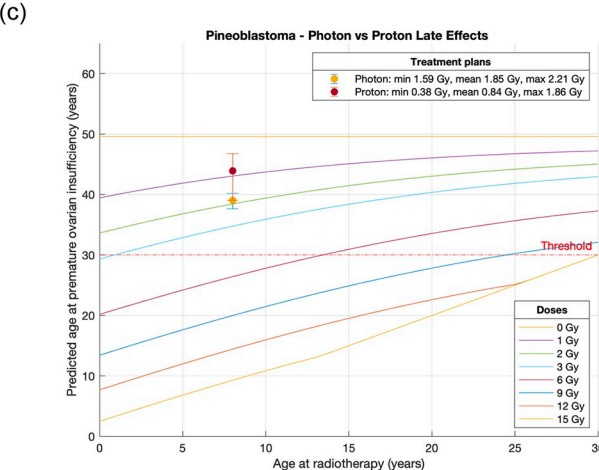

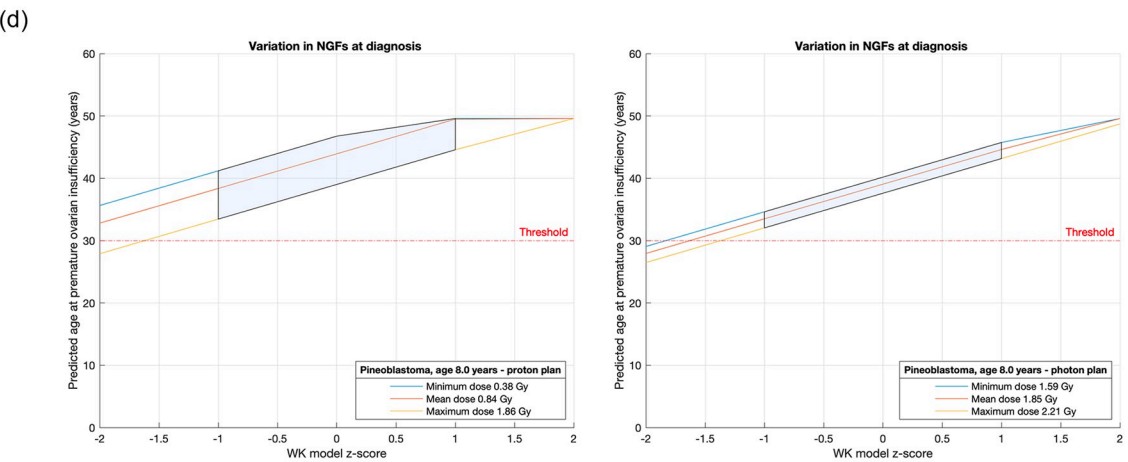

**Fig 4. a**. Treatment plans for Case 1, an 8-year-old with pineoblastoma. **b**. Analysis and comparison of dose received by each ovary for Case 1, an 8-year-old with pineoblastoma. **c**. Comparison of treatment plans assuming average NGF population at age of treatment (i.e., z-score = 0) for Case 1, an 8-year-old with pineoblastoma. Dots are predicted age at menopause after mean dose. Fertility window ranges between predicted ages for minimum & maximum dose. $LD_{50}$ is estimated to be 2 Gy. Dose is to the least affected ovary. **d**. Comparison of treatment plans for Case 1, an 8-year-old with pineoblastoma that incorporate uncertainty due to variation in NGF population at birth. (Or, equivalently, variation in ages at menopause). The z-score 0 value are the same ranges as reported in Fig 6. 68% of the population are expected to fall in the left-to-right shaded area.

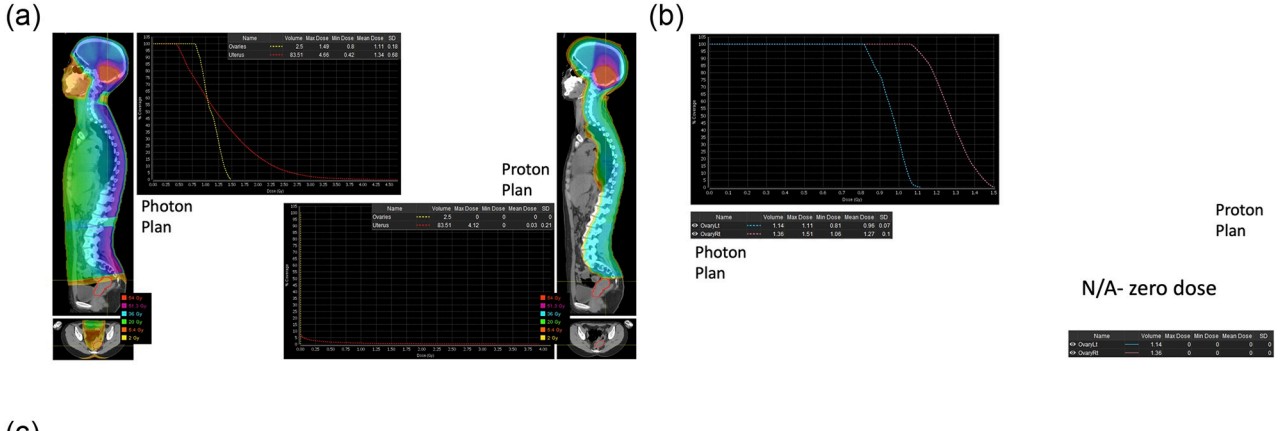

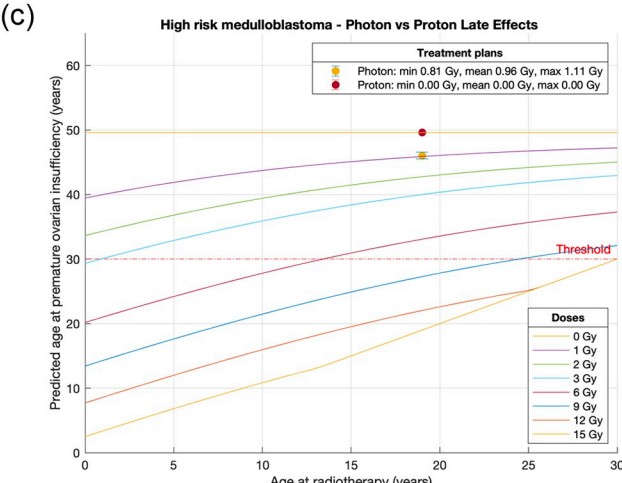

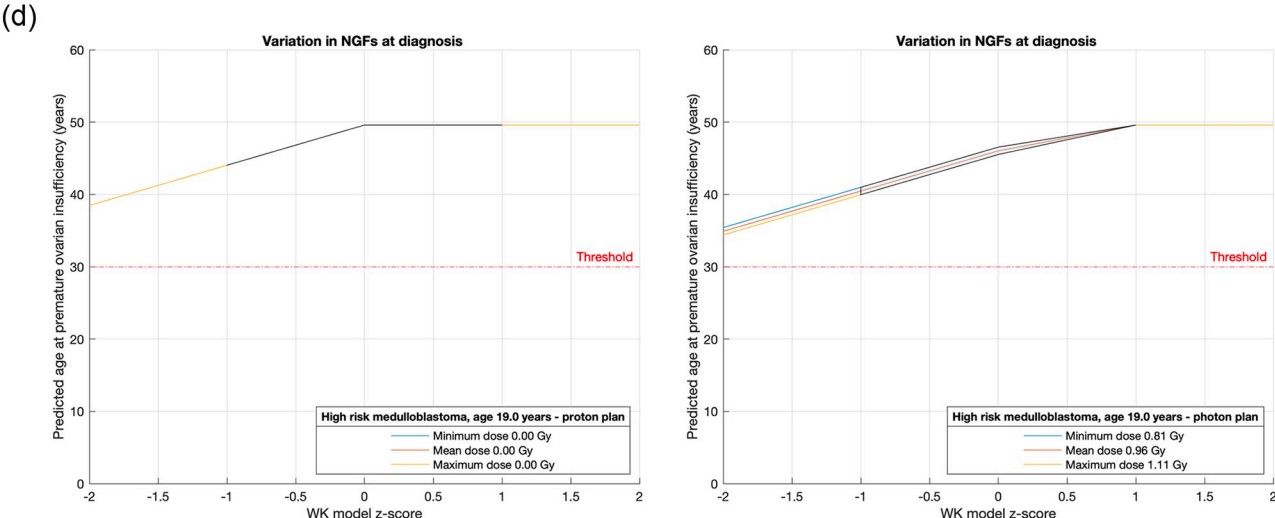

**Fig 5. a**. Treatment plans for Case 2, a 19-year-old with high risk medulloblastoma. **b**. Analysis and comparison of dose received by each ovary for Case 2, a 19-year-old with high risk medulloblastoma. **c**. Comparison of treatment plans assuming average NGF population at age of treatment (i.e., z-score = 0) for Case 2, a 19-year-old with high risk medulloblastoma. Dots are predicted age at menopause after mean dose. Fertility window ranges between predicted ages for minimum & maximum dose. $LD_{50}$ is estimated to be 2 Gy. Dose is to the least affected ovary. **d**. Comparison of treatment plans for Case 2, a 19-year-old with high risk medulloblastoma that incorporate uncertainty due to variation in NGF population at birth. (Or, equivalently, variation in ages at menopause). The z-score 0 values are the same ranges as reported in Fig 6. 68% of the population are expected to fall in the left-to-right shaded area.

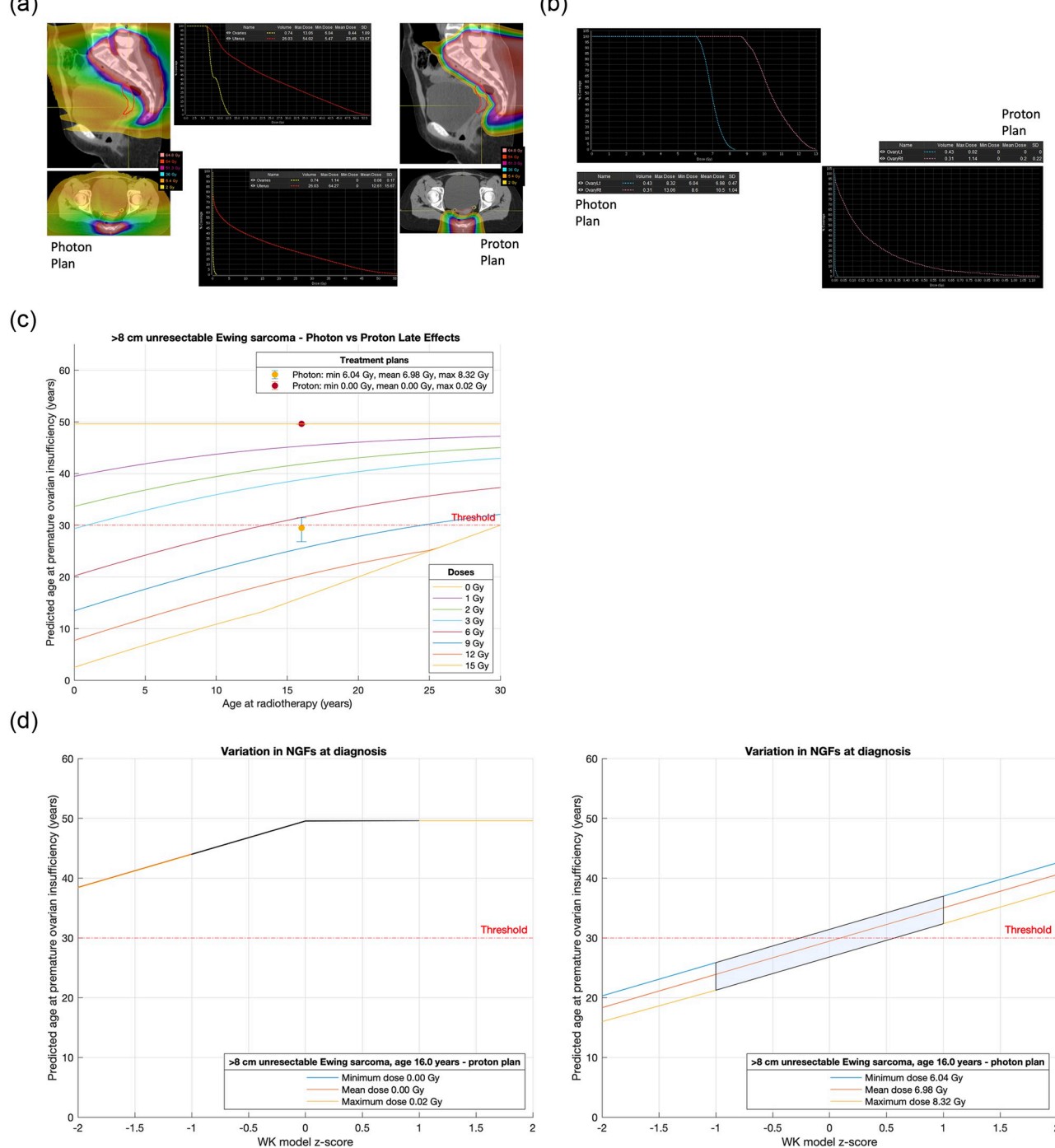

**Fig 6. a**. Treatment plans for Case 3, a 16-year-old with a >8 cm unresectable Ewing sarcoma. **b**. Analysis and comparison of dose received by each ovary for Case 3, a 16-year-old with a >8 cm unresectable Ewing sarcoma. **c**. Comparison of treatment plans assuming average NGF population at age of treatment (i.e., z-score = 0) for Case 3, a 16-year-old with a >8 cm unresectable Ewing sarcoma. Dots are predicted age at menopause after mean dose. Fertility window ranges between predicted ages for minimum & maximum dose. $LD_{50}$ is estimated to be 2 Gy. Dose is to the least affected ovary. **d**. Comparison of treatment plans for Case 3, a 16-year-old with a >8 cm unresectable Ewing sarcoma that incorporate uncertainty due to variation in NGF population at birth. (Or, equivalently, variation in ages at menopause). The z-score 0 values are the same ranges as reported in Fig 6. 68% of the population are expected to fall in the left-to-right shaded area.

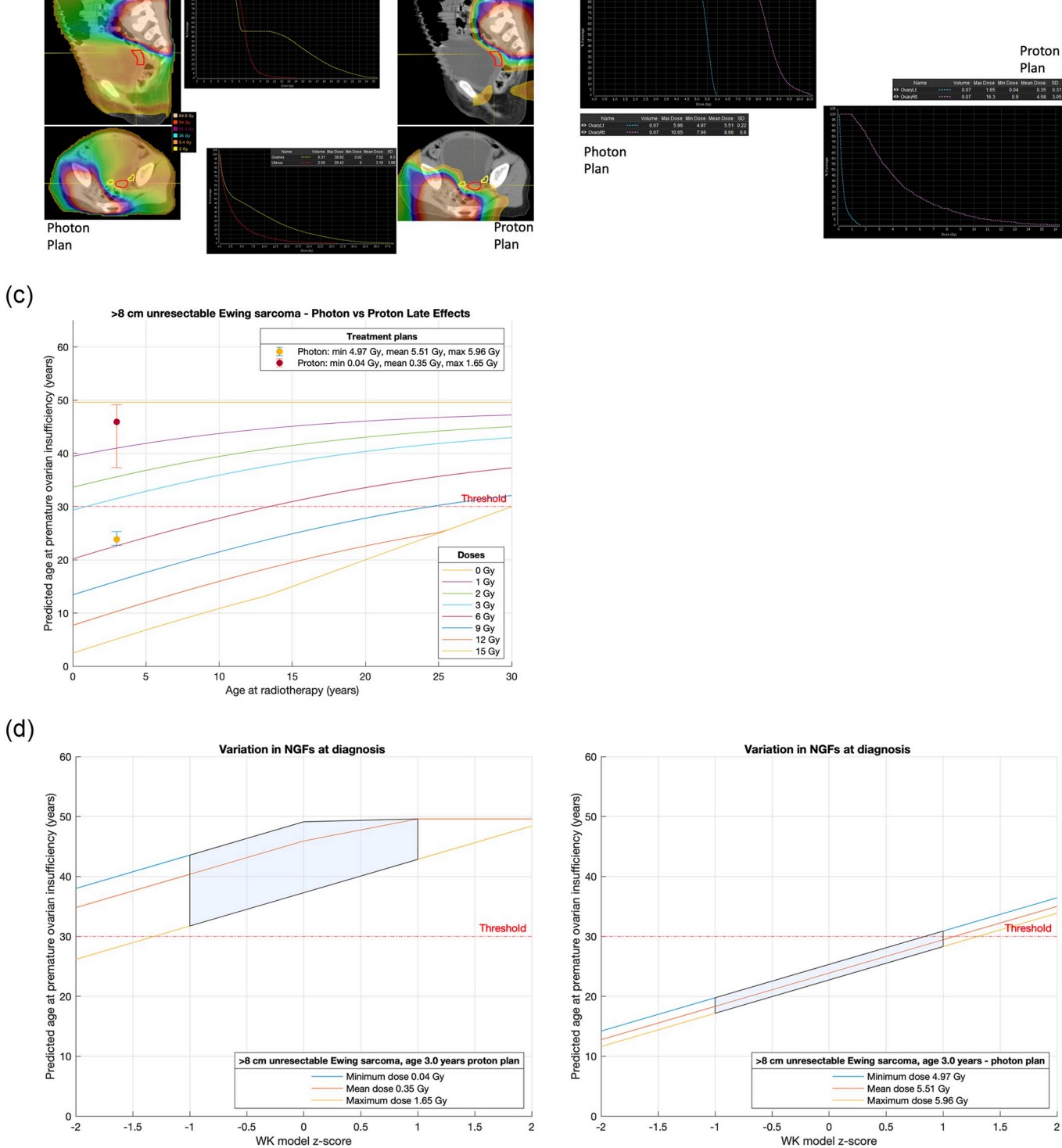

**Fig 7. a**. Treatment plans for Case 4, a 3-year-old with a >8 cm unresectable Ewing sarcoma. **b**. Analysis and comparison of dose received by each ovary for Case 4, a 3-year-old with a >8 cm unresectable Ewing sarcoma. **c**. Comparison of treatment plans assuming average NGF population at age of treatment (i.e., z-score = 0) for Case 4, a 3-year-old with a >8 cm unresectable Ewing sarcoma. Dots are predicted age at menopause after mean dose. Fertility window ranges between predicted ages for minimum & maximum dose. $LD_{50}$ is estimated to be 2 Gy. Dose is to the least affected ovary. **d**. Comparison of treatment plans for Case 4, a 3-year-old with a >8 cm unresectable Ewing sarcoma that incorporate uncertainty due to variation in NGF population at birth. (Or, equivalently,

variation in ages at menopause). The z-score 0 values are the same ranges as reported in Fig 6. 68% of the population are expected to fall in the left-to-right shaded area.

series of two charts. For practical counselling purposes we can assume that if POI is predicted before the age of 30 years (threshold as illustrated) the patient is at high risk of not having a natural window of opportunity for fertility.

Case 1 is an 8-year-old with pineoblastoma. The treatment plans contrast photon and proton techniques and analysis of dose received by the ovary furthest away from the radiation target are shown in Fig 4a and 4b; the plans are compared assuming zero z-score for NGF number at age of treatment (Fig 4c) and compared incorporating the natural variation in NGF number in terms of z-score uncertainty for NGF number (Fig 4d). In both Fig 4c and 4d we have drawn a theoretical threshold for POI at 30 years. In case 1 there may be a small advantage for proton radiotherapy (mean age at POI in the average case 44 years versus 39 years for photon technique) but POI is not expected to occur before 30 years of age in either treatment scenario.

Case 2 is a 19-year-old with high risk medulloblastoma requiring craniospinal irradiation (Fig 5a and 5b). With a proton technique the doses to the ovaries are negligible whereas with a photon technique the expected age at POI in the average case is 46 years (Fig 5c and 5d). The unquantifiable effect of chemotherapy on ovarian function will very likely be additive and further reduce the age at which POI occurs.

Cases 3 and 4 contrast a 16-year-old (Fig 6a and 6b) and a 3-year-old (Fig 7a and 7b) respectively with a >8 cm unresectable pelvic Ewing sarcoma requiring local radiotherapy. Proton and photon plans are made in each case allowing an estimate of the dose to the ovary furthest from the radiation field. For the 16-year-old the proton technique avoids a dose to the ovary whereas the photon plan will compromise ovarian function and predicts POI at or around age 30 years (Fig 6c and 6d). If the only option for this 16-year-old is photon therapy, then fertility preservation may be considered. In the case of the 3-year-old both techniques will compromise ovarian function, but photons are unlikely to give this child any chance of natural fertility with POI supervening in the average case at 24 years as opposed to a proton technique when in the average case POI supervenes at 45 years (Fig 7c and 7d).

## Discussion

Using a validated model of the age-dependent number of NGFs in the human ovary and calculating or estimating the radiation dose received by the ovary furthest away from the high dose target when delivering either photon or proton beams, we have developed an algorithm to estimate the age at which POI is likely to occur. We have taken into account the age-specific decline in the number of primordial follicles in the human ovary, and this allows us to predict the onset of premature ovarian insufficiency in young women before radiotherapy for cancer.

There are several advantages to this mechanistic model of response to ovarian radiation exposure. exposure. Firstly, as we know that ability to conceive a pregnancy declined rapidly within 10 years of the onset of the menopause this will aid clinicians in deciding whether to offer currently available established or experimental fertility preservation fertility preservation techniques are appropriate for an individual patient. Secondly, revision of the radiotherapy plan using either proton or photon radiotherapy will give the clinician the opportunity to improve the plan to avoid or decrease the radiation exposure to the ovary furthest away from the high dose radiation target.

A recent model published by St Jude Children's Research Hospital and the Childhood Cancer Survivor Study (CCSS) group has produced an algorithm that predicts the risk of acute

ovarian failure (AOF) within five years of treatment [22]. Our algorithm improves upon this model by allowing us to predict within well-defined confidence intervals the reproductive window for an individual patient at all ages up to the age of natural (i.e., untreated) menopause.

The advent of proton radiotherapy has allowed clinicians to devise treatment plans that will decrease the dose exposure to organs at risk close to the radiation target. By comparing proton and photon radiotherapy treatment plans the clinician can with reasonable accuracy determine the safest treatment plan for an individual patient and provide a reasonable estimate of the age at which premature ovarian insufficiency will supervene. With the rapid development of both established and experimental techniques of fertility preservation for younger patients, including ovarian tissue cryopreservation [23] and ovarian transposition [24], selection of patients for fertility preservation should be evidence-based. Our online algorithm, which predicts age at POI (and not AOF within 5 years as returned by the St Jude/CCSS online tool [22]) improves scope for both accurate fertility prediction, and potential justification for resource-intensive (and less readily available) fertility preservation techniques.

There are several limitations to our algorithm. Probably the most important is the unknown additional and unquantifiable effect of chemotherapy for each individual patient. Inevitably, chemotherapy which is part of most treatment plans for young patients with cancer will be additive in an unquantifiable way to the prediction of when POI is likely to occur. An important strength of our methods is accurate and validated prediction of ovarian reserve and age at menopause in the base case of zero radiotherapy dose. Our model predicts average age at menopause to be 49.6 years, in close agreement with reports from prospective cohorts: 50.4 years (SD 3.9 years) [25] and 50.1 years (SD 4.1 years) [26]. When combined with an age-related model for ovarian volume [27], our model has high predictive accuracy for mean follicle density in healthy subjects for ages 16–37 years [28] and has been used to assess fertility preservation options in young women with Turner syndrome [29] and after first-line cancer treatment for adult women [30] and girls under 18 years of age [31].

The four examples illustrate that, for a selection of diagnoses and treatment plans, proton radiotherapy may hold advantages for the individual patient in terms of the dose received by the ovary furthest away from the treatment volume and hence the age at which POI may occur. In cases three and four representing the 16-year-old and the 3-year-old respectively with an unresectable pelvic Ewing sarcoma there appears to be a clear advantage for proton radiotherapy in both cases but in the other examples the magnitude of the advantage may be less pronounced.

With accessible user-friendly tools, assessment of the likely reproductive lifespan for an individual female patient with a curable cancer should now become part of the standard of care. Our online tool enables a reasonable prediction of reproductive lifespan with the main caveat being that the effects of chemotherapy will be additive and therefore our prediction of fertility window is an upper limit for the individual patient. The incorporation of chemotherapy represents the obvious next step for model refinement. Additionally, dose modeling of other organs in the female reproductive tract (uterus, cervix) as well as the male gonads might benefit from similar online tools to move toward precision medicine.

## Author Contributions

**Conceptualization:** Thomas W. Kelsey, Chia-Ho Hua, Danny Indelicato, W. Hamish Wallace.

**Data curation:** Thomas W. Kelsey, Amber Wyatt, Danny Indelicato.

**Formal analysis:** Thomas W. Kelsey, Chia-Ho Hua, Amber Wyatt, Danny Indelicato, W. Hamish Wallace.

**Investigation:** Thomas W. Kelsey, Chia-Ho Hua, Amber Wyatt, Danny Indelicato, W. Hamish Wallace.

**Methodology:** Thomas W. Kelsey, Chia-Ho Hua, Amber Wyatt, Danny Indelicato, W. Hamish Wallace.

**Project administration:** Thomas W. Kelsey.

**Software:** Thomas W. Kelsey, Chia-Ho Hua.

**Supervision:** W. Hamish Wallace.

**Validation:** Thomas W. Kelsey, Chia-Ho Hua, Amber Wyatt, Danny Indelicato, W. Hamish Wallace.

**Visualization:** Thomas W. Kelsey, Chia-Ho Hua, Amber Wyatt, Danny Indelicato.

**Writing – original draft:** Thomas W. Kelsey, Danny Indelicato, W. Hamish Wallace.

**Writing – review & editing:** Thomas W. Kelsey, Chia-Ho Hua, Danny Indelicato, W. Hamish Wallace.

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
