## [Decision Letter · Decision Letter 0]

11 Sep 2022

PONE-D-22-15540A predictive model of the effect of therapeutic radiation on the human ovaryPLOS ONE

Dear Dr. Kelsey,

Thank you for submitting your manuscript to PLOS ONE. After careful consideration, we feel that it has merit but does not fully meet PLOS ONE’s publication criteria as it currently stands. Therefore, we invite you to submit a revised version of the manuscript that addresses the points raised during the review process. The reviewers are critical of its presentation. There are several criticisms that must be adequately addressed. These require a detailed response and may require additional data or experiments. .

We look forward to receiving your revised manuscript.

Kind regards,

Meijia Zhang

Academic Editor

PLOS ONE

Journal Requirements:

Additional Editor Comments:

The reviewers were critical of its presentation. There are several criticisms that must be adequately addressed. These require a detailed response and may require additional data or experiments.

Reviewers' comments:

Reviewer's Responses to Questions

**Comments to the Author**

1. Is the manuscript technically sound, and do the data support the conclusions?

Reviewer #1: Yes

Reviewer #2: Yes

2. Has the statistical analysis been performed appropriately and rigorously? 

Reviewer #1: Yes

Reviewer #2: Yes

3. Have the authors made all data underlying the findings in their manuscript fully available?

Reviewer #1: Yes

Reviewer #2: Yes

4. Is the manuscript presented in an intelligible fashion and written in standard English?

Reviewer #1: Yes

Reviewer #2: Yes

5. Review Comments to the Author

Reviewer #1: 1.The authors should give a specific definition of the non-growing follicle(NGF).

2.Line 172 and 173: the reference was outdated.

3.Figure 4a: the line chart of the proton plan lacked a Y-axis title.

Reviewer #2: The POI caused by radiotherapy for cancer in young women has a significant impact on women's fertility. This study provides a model for predicting the age of occurrence of POI and allows detailed predictions of the impact of various radiotherapy plans on fertility, which helps to understand the fertility risk of patients more clearly.

1 The four cases in this study are fewer, the coverage should be more comprehensive.

2 There seems to be a problem in the description of the method, for example, lines 134-146 seem to be in the result.

3 It is confused that the description of fig.1, fig.2 and fig.3 appear in methods.

4 The results of fig.4-fig.7 are too scattered and should be integrated. Why are the figures separate from each other in fig.4abcd? Also include fig.5 abcd, fig. 6abcd, fig.7abcd?

5 The structure of this manuscript is a little confused.

6. PLOS authors have the option to publish the peer review history of their article (what does this mean?). If published, this will include your full peer review and any attached files.

Reviewer #1: No

Reviewer #2: No

---

## [Author Response · Author response to Decision Letter 0]

23 Sep 2022

Dear Meijia Zhang Academic Editor PLOS ONE,

Thank you for your detailed and helpful reviews of our manuscript.

We have carefully considered PLOS ONE’S style requirements and made any changes that were necessary. See revised manuscript.

2. In your Data Availability statement, you have not specified where the minimal data set underlying the results described in your manuscript can be found. PLOS defines a study's minimal data set as the underlying data used to reach the conclusions drawn in the manuscript and any additional data required to replicate the reported study findings in their entirety. All PLOS journals require that the minimal data set be made fully available. Upon re-submitting your revised manuscript, please upload your study’s minimal underlying data set as either Supporting Information files or to a stable, public repository and include the relevant URLs, DOIs, or accession numbers within your revised cover letter.

The minimal data set underlying our results is the solution to the Wallace-Kelsey model which is publicly available in Table form in the java script directory of the web page. We have added a data availability statement in the manuscript specifically so that the reader can if required source the original data set.

Reviewer 1 and 2 specific comments:

1. Is the manuscript technically sound, and do the data support the conclusions? 

Reviewer #1: Yes

Reviewer #2: Yes

Thank you for these positive assessments.

2. Has the statistical analysis been performed appropriately and rigorously? 

Reviewer #1: Yes

Reviewer #2: Yes

Thank you for these positive assessments.

3. Have the authors made all data underlying the findings in their manuscript fully available?  The PLOS Data policy requires authors to make all data underlying the findings described in their manuscript fully available without restriction, with rare exception (please refer to the Data Availability Statement in the manuscript PDF file). The data should be provided as part of the manuscript or its supporting information, or deposited to a public repository. For example, in addition to summary statistics, the data points behind means, medians and variance measures should be available. If there are restrictions on publicly sharing data—e.g. participant privacy or use of data from a third party—those must be specified.

Reviewer #1: Yes

Reviewer #2: Yes

Thank you for these positive assessments. See our response above regarding public availability of the data set.

4. Is the manuscript presented in an intelligible fashion and written in standard English?  PLOS ONE does not copyedit accepted manuscripts, so the language in submitted articles must be clear, correct, and unambiguous. Any typographical or grammatical errors should be corrected at revision, so please note any specific errors here.

Reviewer #1: Yes

Reviewer #2: Yes

Thank you for these positive assessments.

5. Review Comments to the Author  

Reviewer #1: 1.The authors should give a specific definition of the non-growing follicle(NGF). 

An accepted definition of the non-growing human follicle is as follows: Non growing follicles (NGF) contain immature oocytes surrounded by flat, squamous granulosa cells (support cells) that are segregated from the oocyte's environment by the basal lamina. They are quiescent, showing little to no biological activity. They represent the human ovarian reserve. We have inserted this definition into the introduction and standardised our terminology throughout the manuscript using NGF throughout.

2.Line 172 and 173: the reference was outdated.

Thank you. We have updated the reference in relation to the assessment of ovarian reserve in those women under 25 years to PMID 32315387, published in 2020.

3.Figure 4a: the line chart of the proton plan lacked a Y-axis title.

Thank you. We have corrected this error. The y-axis is % coverage as in the other figures.

Reviewer #2: The POI caused by radiotherapy for cancer in young women has a significant impact on women's fertility. This study provides a model for predicting the age of occurrence of POI and allows detailed predictions of the impact of various radiotherapy plans on fertility, which helps to understand the fertility risk of patients more clearly.

Thank you for these positive comments.

1. The four cases in this study are fewer, the coverage should be more comprehensive.

The four cases are exemplars. The aim of the four cases is to demonstrate the potential application of using our algorithm to predict with confidence intervals the age at POI. This is useful for counselling and for discussions around fertility preservation. We think more cases will not add anything to the manuscript as they illustrate the methodology in four representative patients and are not the results of our manuscript.

2 There seems to be a problem in the description of the method, for example, lines 134-146 seem to be in the result.

This comment refers to our methodology for the models that underpin the algorithm as shown in figures 2 and 3. We are happy to accommodate this referee’s comment and have transferred the section that describes Figures 2 and 3 (lines 134-146) to the results section.

3 It is confused that the description of fig.1, fig.2 and fig.3 appear in methods.

We have, as above, transferred the description of figures 2 and 3 from the methods to the results section. 

4 The results of fig.4-fig.7 are too scattered and should be integrated. Why are the figures separate from each other in fig.4abcd? Also include fig.5 abcd, fig. 6abcd, fig.7abcd?

Each of the figures 4 through figure 7 represent patient exemplars with each example having four sub figures. We do not disagree with the sentiment of the reviewer, but if they are combined into a single figure with four panels we cannot retain the fine detail and retain image publication quality required by your journal. 

5 The structure of this manuscript is a little confused

We hope that by moving the explanation describing figures 2 and 3 into the results section the structure of the manuscript is easier to follow.

---

## [Decision Letter · Decision Letter 1]

19 Oct 2022

A predictive model of the effect of therapeutic radiation on the human ovary

PONE-D-22-15540R1

Dear Dr. Kelsey,

We’re pleased to inform you that your manuscript has been judged scientifically suitable for publication and will be formally accepted for publication once it meets all outstanding technical requirements.

Kind regards,

Meijia Zhang

Academic Editor

PLOS ONE

Additional Editor Comments (optional):

Reviewers' comments:

Reviewer's Responses to Questions

**Comments to the Author**

1. If the authors have adequately addressed your comments raised in a previous round of review and you feel that this manuscript is now acceptable for publication, you may indicate that here to bypass the “Comments to the Author” section, enter your conflict of interest statement in the “Confidential to Editor” section, and submit your "Accept" recommendation.

Reviewer #1: All comments have been addressed

Reviewer #2: All comments have been addressed

2. Is the manuscript technically sound, and do the data support the conclusions?

Reviewer #1: Yes

Reviewer #2: Yes

3. Has the statistical analysis been performed appropriately and rigorously? 

Reviewer #1: Yes

Reviewer #2: Yes

4. Have the authors made all data underlying the findings in their manuscript fully available?

Reviewer #1: Yes

Reviewer #2: Yes

5. Is the manuscript presented in an intelligible fashion and written in standard English?

Reviewer #1: Yes

Reviewer #2: Yes

6. Review Comments to the Author

Reviewer #1: (No Response)

Reviewer #2: (No Response)

7. PLOS authors have the option to publish the peer review history of their article (what does this mean?). If published, this will include your full peer review and any attached files.

Reviewer #1: No

Reviewer #2: No

---

## [Editor Report · Acceptance letter]

9 Nov 2022

PONE-D-22-15540R1 

A predictive model of the effect of therapeutic radiation on the human ovary 

Dear Dr. Kelsey:

I'm pleased to inform you that your manuscript has been deemed suitable for publication in PLOS ONE. Congratulations! Your manuscript is now with our production department. 

Kind regards, 

on behalf of

Dr. Meijia Zhang 

Academic Editor

PLOS ONE